# The Role of Starch in Shaping the Rheo-Mechanical Properties of Fat-in-Water Emulsions

Ryszard Rezler ⬤

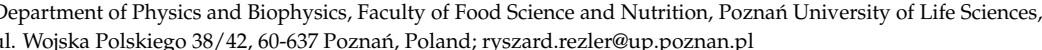

Department of Physics and Biophysics, Faculty of Food Science and Nutrition, Poznań University of Life Sciences, ul. Wojska Polskiego 38/42, 60-637 Poznań, Poland; ryszard.rezler@up.poznan.pl

**Abstract:** The DMA technique was used to conduct experiments on the rheo-mechanical properties of emulsified bovine fat meat products stabilised with potato starch. Starch gels with starch concentrations corresponding to the concentration of starch in water in the emulsions under analysis were used as control systems. The research showed that the rheo-mechanical properties of starch gels and starch–fat gels result from the conformational changes occurring within the structural elements of their spatial network. In starch gels, segments formed by complex associations of amylose chains are structural elements, whereas in starch–fat gels (emulsions) these are additionally amylose–fat complexes. Changes occurring during progressive retrogradation increase the degree of cross-linking in them. In starch gels, they are conditioned by the starch concentration, whereas in emulsions they are conditioned by the concentration of starch and the presence of fat. The parameters obtained by adjusting the Avrami equation to the data obtained with the DMA method enabled the determination of three forms of organisation of the dispersion structure of starch–fat systems. Each of these forms of structure organisation is conditioned by the concentration of starch in the emulsion system.

**Keywords:** complex index; emulsion; bovine fat; potato starch; rheology

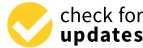



## 1. Introduction

Emulsions are colloidal dispersions which consist of at least two immiscible liquids (usually water and oil), one of which is dispersed as droplets in the other [1]. In general, there are two types of emulsions: water dispersed in oil (w/o) and oil dispersed in water (o/w). Both of them are used in the technological processes of various emulsified food products. However, oil-in-water (o/w) emulsion systems are more common. Emulsions can also be divided according to their function in food. There are emulsions which are the final product in a technological process, e.g., in the production of milk, cream liqueurs, dressings, ice cream. There are also emulsions which form more complex structures in multicomponent products, e.g., in yoghurts and other systems in the form of a gel, as well as in cold cuts (sausages, pâtés). The nature of the emulsion gives food products various functional characteristics such as the right appearance, texture, mouthfeel, and flavour profiles [2–5]. Moreover, emulsions are widely used as carriers for capsules delivering bioactive agents such as vitamins and nutraceuticals. The properties of emulsions can be largely changed by selecting a different oil or a different type of emulsifier, by including various additives (such as thickeners or gelling agents), by varying the distribution of droplets of different sizes and by manipulating the oil-to-water ratio. As a result, emulsions with different physicochemical, sensory, and nutritional properties can be obtained. However, there are some limitations to the use of emulsions in the food industry. These limitations result from the fact that such dispersion systems tend to become thermodynamically destabilised as a result of such processes as flocculation and coalescence. In consequence, emulsions become physically unstable [6]. Moreover, they have a limited ability to control the profile of releasing encapsulated ingredients. Apart from that, high fat content is often necessary to form an oil-in-water emulsion with a viscous or semi-solid texture. This is a disadvantage

when low-calorie versions of some food products are prepared, especially due to the fact that recently there has been growing interest in improving the health-promoting properties of food, its shelf life, and safety. These new products are being developed in order to cope with some negative effects of modern food production technologies, such as the increasing incidence of chronic diseases, environmental damage, and food safety concerns. For this reason, scientists are searching for formulas of emulsion systems with new or improved functional attributes, such as lower caloric content [7], controlled digestion behaviour [8], and increased bioavailability of bioactive compounds [9]. However, when the fat content in an emulsion-based food product is reduced, it usually adversely affects its desirable sensory and physicochemical properties [10]. It is not easy to develop formulations of new assortments of products of lower calorific value and simultaneously maintain the physico-chemical parameters and, in particular, the structural traits desired by consumers [10]. A number of commonly consumed emulsified food products consist of mixtures of different kinds of colloidal particles. If a small amount of fat has been replaced and the fatty phase constitutes an oil-in-water emulsion, proteins found in the system are capable of binding water without significant deterioration of the product quality [11]. When more than 70% of fat has been replaced, which is typical of low-fat products, the continuous phase of the product is transformed into an oil-in-water emulsion and simultaneously loses the rheo-mechanical characteristics of condensed emulsions [12,13]. This leads to considerable changes in the quality parameters of the product, especially in its texture and water-binding capacity [14]. Fat is replaced by water with other additives which not only bind water but also have structure-forming properties. Such features are particularly characteristic of hydrocolloid preparations based on proteins, starch as well as other polysaccharides [15,16]. Among all polysaccharides, starch, especially its amylose fraction, is one of the most important structure-forming food constituents due to its physicochemical properties. This polysaccharide can form inclusion complexes with both organic and inorganic compounds [17]. As a result, the physicochemical properties of starch (e.g., water solubility and retrogradation) can be modified [18].

Therefore, many researchers conducting experiments focus on the gelation and retrogradation of starch in the presence of fats [19,20]. Most of these studies concern monoglycerides or individual fatty acids.

The authors of most studies on the influence of fats on the properties of the starch system conduct experiments on vegetable fats. Few researchers have investigated the relationship between starch systems and animal fats, because the structure of fatty acids of animal origin is more complex than the structure of fatty acids of plant origin. This applies not only to the content of saturated fatty acids, which predominate in animal fats, but also to their molecular structure. If researchers conduct such studies, most of them focus on the stabilisation of the structure of the continuous phase in low-fat meat products and only test the properties of the final product [21,22].

So far there have been few studies in which researchers explained how the partial supplementation of fat with starch in low-fat meat products influenced the molecular conditions of their mechanical and textural properties.

For this reason, research was conducted to determine how the replacement of fat with native starch in animal (bovine) fat-in-water emulsions, which were used as models of the continuous phase in low-fat meat products, influenced their rheological properties.

The aims of the study were to:

- Investigate the influence of using native starch as a stabiliser of the animal fat (beef)-in-water emulsion on its molecular structure;
- Determine the influence of changes occurring in the structural elements of emulsions resulting from starch retrogradation on their rheo-mechanical properties;
- Determine the influence of the amount of the starch additive on the stability of the emulsions.

Obtaining full information on the molecular conditions of the formation of the structure of animal fat-in-water emulsions stabilised with starch as well as the information on

how structural changes influence the rheological properties of these emulsions will help to control the functionality of meat products, improve their texture, and help to develop new strategies for producing sausages with a healthier profile (low-fat meat products with a healthier lipid composition).

## 2. Materials and Methods

### 2.1. Materials and Sample Preparation

Experiments were conducted on fat-in-water emulsions with a fat-to-water ratio of 1:3 (g). The emulsions were supplemented with unmodified potato starch (made by Trzemeszno, Poland) added in the following quantities: 1.0, 0.8, 0.6, 0.5, 0.4, and 0.2 of fat mass, which corresponds to starch concentrations of 0.2, 0.17, 0.13, 0.11, 0.09, and 0.05 g/g, respectively. The starch concentration in the emulsion was defined as $m_s/(m_s + m_w + m_f)$, where: $m_s$—starch mass, $m_w$—water mass, $m_f$—fat mass. For comparison, tests were conducted on starch gels in which the starch content in water was analogous to the starch content in the emulsions.

The chemical composition of the starch was as follows: moisture—13.2%, protein—0.2%, fat—0.1%, ash—0.35%, amylose—25%. The iodine-binding procedure [23] with spectrophotometric detection was applied to measure the amylose content (Shimadzu 2001).

Liquid bovine fat (Morliny, Poland) was added to the experimental system at a temperature of 40 °C. Emulsion samples weighing 100 g underwent thermal treatment at a constant temperature of 75 °C in an LWC2M water bath (DANLAB Company, Switzerland) for 1 h. The samples were stirred continuously in an Ika Eurostar Power Control-Visc 6000 agitator (Ika Werke GmbH, Staufen, Germany).

### 2.2. Rheo-Mechanical Properties

A DMWT dynamic mechanical analysis (DMA) spectrometer (Cobrabid, Poznań, Poland) was used for measurements. In free-vibration rheometers, the viscoelastic properties of systems are calculated by analysing the parameters characterising the curve of the pendulum-free, damped vibrations with and without the sample (vibration frequency and damping decrement). A cone-plate ($\phi = 0.03$ m, $\alpha = 6°$) measuring system was applied. The component of the complex modulus of elasticity G′ (storage modulus) and the loss tangent (tgδ) were determined. G′ is associated with this part of potential deformation energy which is maintained in the course of periodical deformations. The loss tangent (tgδ) is a measure of internal friction. Its value shows the relative quantity of energy dissipated in the material in the course of one deformation cycle. The system vibration frequency was 1.2 Hz. The starch structuration kinetics were measured at a set temperature of 25 °C (accuracy ± 0.1 °C). The linear viscoelastic region in each sample was taken into consideration in the analyses.

### 2.3. Composition of Fatty Acids

The composition of fatty acids was assessed with the GLC method in a Hewlett-Packard 5890 SH apparatus with a Supelcowax capillary column (30 m × 0.25 mm − 0.25 μm) and a flame ionisation detector (FID). The analysis was conducted at a programmed temperature ranging from 60 °C to 200 °C and a temperature increase of 12 °C/min. The final temperature was maintained for 10 min. The detector and injector worked at 240 °C. The measurement error was 5% and the detection limit was 0.1% [24].

### 2.4. Amylose Content and Complex Formation

The capability of formation of the starch–iodine complex (CI) was investigated with the method described in the study by Biais et al. [25]. Measurements were conducted on gelatinised starch and bovine fat mixtures. The gelatinised starch and water mixtures were used as reference samples. The gelatinised starch and fat systems as well as the starch and water systems were prepared according to the procedure described in Section 2.1.

The absorbance (ABS) of the starch and fat mixtures as well as the starch and water mixtures were measured with a Shimadzu 2001 spectrophotometer at 690 nm. The CI was calculated with the following formula: CI, % = 100x (ABS of reference − ABS of sample)/ABS of reference. The measurements were triplicated, and the results were averaged.

### 2.5. Emulsion Stability

The centrifuge stability (thermostability) of model emulsions was assessed by centrifuging the samples (MPW-352 centrifuge, MPW Med. Instruments, Warsaw, Poland). Samples 10 cm$^3$ in size of the fat-in-water emulsion with starch were placed in test tubes, closed with parafilm and left for 24 h at room temperature. Next, the emulsions were heated at 75 ± 1 °C in a SUP-100M drying oven (ZALIMP, Warsaw, Poland) for 5 min and centrifuged at 3000 rpm for 5 min. The ratio of the volume of the centrifuged/delaminated sample to the total volume of the sample before centrifugation was used as a measure of the stability of the systems under analysis. If no phase separation was observed, the stability of the emulsion was 100% [26]. The volume of the centrifuged layer was determined by measuring the thickness of the layer and the diameter of the test tube. A cathetometer (Eberbach's E5163.00, Belleville, MI, USA) and a calliper were used for the measurements.

The stability of the emulsion was expressed as a percentage according to the following formula:

$$SE = \frac{V_m - V_w}{V_m} \, 100\% \tag{1}$$

$$v_w = v_o + v_{H_2O} \tag{2}$$

where:

$v_m$—the volume of the emulsion under analysis [cm$^3$];

$v_w$—the volume of centrifuged oil and water [cm$^3$];

$v_o$—the volume of oil;

$v_{H_2O}$—the volume of water.

The percentage shares of individual fractions in the centrifuged/stratified emulsion sample were also calculated.

### 2.6. Statistical Analysis

The mean and standard deviations were calculated on the basis of the rheo-mechanical and emulsion stability measurements of three newly prepared samples. The Microsoft Excel 2011 package was used for statistical analyses. The values of changes in the k, f, and m parameters were calculated by computer fitting (least squares method, TableCurve 2D).

## 3. Results

In order to characterise the rheo-mechanical traits of the experimental starch gels and starch–fat emulsions, changes in the kinetics of the basic parameters determining their rheo-mechanical properties were analysed. The results of the dynamic mechanical analysis of the starch systems over time counted from the moment of reaching the ambient temperature at different starch concentrations can be seen in the course of the dependence of storage modulus $G'(t)$ shown in Figure 1.

The initial values different from zero of the dynamic rigidity modulus $G_{o1}$ (on average for gels: 0.14 g/g < cs ≤ 0.25 g/g ÷ 800 Pa, $c_s$ = 0.14 g/g ÷ 440 Pa, and $c_s$ < 0.14 g/g ÷ 240 Pa and emulsions: 0.13 g/g < $c_s$ ≤ 0.20 g/g ÷ 900 Pa, $c_s$ = 0.13 g/g ÷ 500 Pa, and $c_s$ < 0.13 g/g ÷ 260 Pa) indicate that already during the cooling process a spatial network developed in the examined systems with the concentration of $n_{os}$ ($c_s$) segments dependent on the concentration of starch. The approximate correlation between the storage modulus of highly elastic polymer networks and the concentration of segments in these networks was determined according to the following formula [22]:

$$n_s \cong \frac{G'}{RT} \tag{3}$$

where:

   $n_s$—concentrations of network segments, G′—storage modulus, R—gas constant.

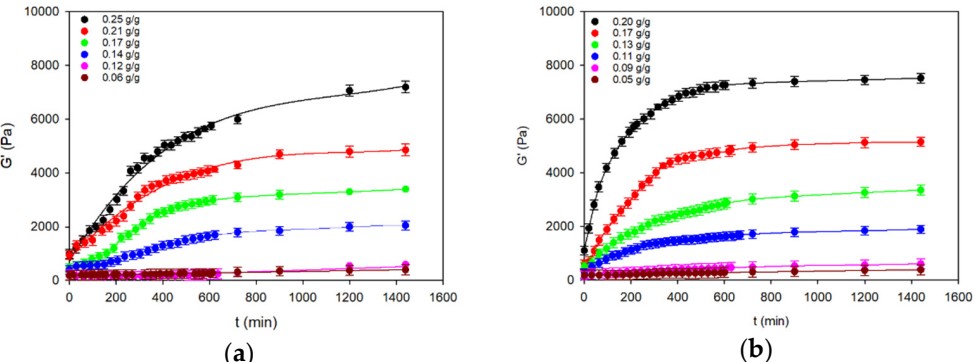

**Figure 1.** The kinetics of the storage modulus (G′) in the process of formation of starch gels (**a**) and starch–fat emulsions (**b**). The values are expressed as means ± SD (*n* = 3).

As the time of cross-linking passed, the storage modulus of hydrogels and emulsions tended to increase at different starch concentrations. This means that the concentration of effective segments in this network was increasing.

It is also noteworthy that the kinetics of the course of the crosslinking processes conforms to the description of the Avrami equations [27]:

$$n_s = n_{os} + [n_{ws} - n_{os}]\{1 - \exp[-(kt)^m]\} \quad (4)$$

where:

$n_{os}$ and $n_{ws}$—the initial and final concentrations of the network segments, respectively;
k—the kinetics constant;
m—the power exponent generally related to crystallite morphology.

After filling Equation (4) with the data obtained with the DMA method, the power exponent of time m assumes values dependent on the starch concentrations in the systems under analysis. In emulsions: m ≅ 1 for 0.13 g/g < $c_s$ ≤ 0.2 g/g, m ≅ 2 for $c_s$ = 0.13 g/g, and m ≅ 3 for $c_s$ < 0.13 g/g, whereas in starch gels with starch concentrations corresponding to the concentration of starch in water in the emulsions: m ≅ 1 at an interval 0.14 g/g < $c_s$ ≤ 0.25 g/g, m ≅ 2 for $c_s$ = 0.14 g/g, and m ≅ 3 for $c_s$ < 0.14 g/g. The course of variation in the kinetic constant *k* (Figure 2) indicates the complexity of the molecular processes leading to the formation of a spatial network in the gel and emulsion systems.

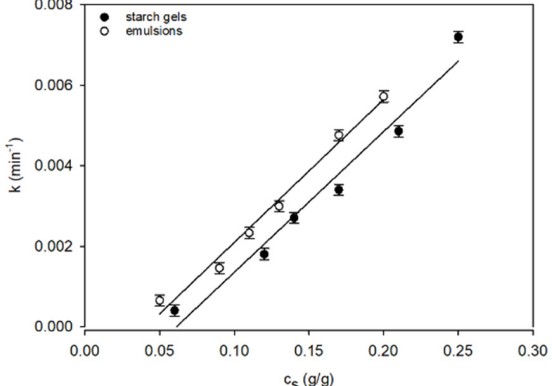

**Figure 2.** The concentration-dependent relationships of the time constant k of the starch gels and starch–fat emulsions. The values are expressed as means ± SD (*n* = 3).

As can be seen in the diagrams in Figure 3, the courses of the equilibrium $n_{ws}$ and initial $n_{os}$ concentrations of the segments of the gel and emulsion network in the squared starch concentration function ($c_s^2$) are linear, with the slope coefficients conditioned by the range of the starch concentration. This means that there are differences in the spatial structures of the starch gels and starch–fat emulsions resulting from starch retrogradation.

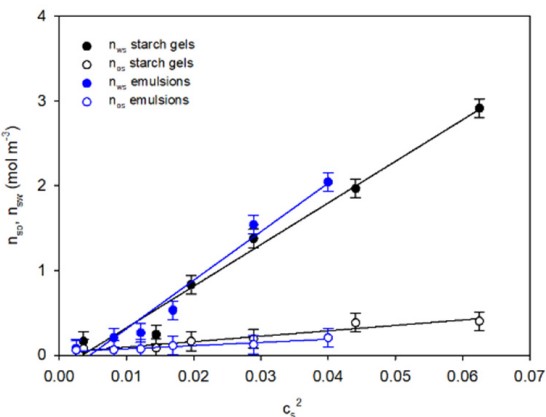

**Figure 3.** The concentration-dependent relationships of initial $n_o$ and final $n_w$ concentrations of spatial network segments of the starch gels and starch–fat emulsions. The values are expressed as means $\pm$ SD ($n = 3$).

As the segment concentration in the gel network is determined by the product of the node concentration (w) and the functionality of these nodes (f):

$$n_s = 0.5\ fw \tag{5}$$

Equation (3) can be presented in the following form:

$$n_s(t) = 0.5\ w\ \{f_o + (f_k - f_o)[1 - \exp[-(kt)^m]]\} \tag{6}$$

where:

$f_o$ and $f_w$—the initial and final functionality of nodes, respectively.

This dependence shows that at a given starch concentration in the system and at a constant temperature the increase in the gel stiffness as a function of time occurs as a result of the expansion of nodes. New macromolecular segments are bound, which increases the average functionality of the set of nodes stabilising the spatial network. Based on the results from the values of the functionalities of the equilibrium network nodes and from the ratio of the directional coefficients of the anamorphoses shown in Figure 3, the initial average functionalities of the network nodes in both systems under analysis assumed comparable values in the entire range of starch concentration $c_s$: for starch gels and emulsions $f_o \cong 7.9$ and 8.0, respectively. The presence of fat did not significantly affect the initial spatial structure of the emulsions.

This means that at the initial stage of retrogradation spatial network nodes in the form of tetrafunctional structures were formed in the systems, which were bispiral associations of amylose chains.

As the gels and emulsions aged, the network underwent gradual reconstruction—the functionality of the initial nodes increased due to the adsorption of successive macromolecular segments on them. This resulted in high mean values of the final functionalities $f_k$ in the starch gels and emulsion systems (Table 1). These high values of the mean final functionalities $f_k$ allow us to assume that the structure of the spatial network of the systems under analysis was determined by the nodes which were complex associations of starch chains (hexagonal associations of bispiral structures) [28,29].

**Table 1.** The dependence of final $f_k$ functionalities on the starch concentration in the starch gels and emulsions. The values are expressed as means $\pm$ SD ($n = 3$).

| Starch Gels | | Emulsions | |
|---|---|---|---|
| $c_s$, g/g | $f_k$ | $c_s$, g/g | $f_k$ |
| 0.25 | $120.0 \pm 4.0$ | 0.20 | $130.1 \pm 6.1$ |
| 0.21 | $94.2 \pm 3.0$ | 0.17 | $112.0 \pm 5.0$ |
| 0.17 | $85.3 \pm 3.0$ | 0.13 | $98.2 \pm 5.0$ |
| 0.14 | $78.0 \pm 2.0$ | 0.11 | $87.3 \pm 4.3$ |
| 0.12 | $68.0 \pm 2.0$ | 0.09 | $72.1 \pm 4.1$ |
| 0.06 | $32.0 \pm 1.5$ | 0.05 | $34.0 \pm 3.0$ |

Earlier, the measurements of the storage modulus $G'$ (Figure 1) showed that a spatial network with different concentrations of segments was formed in the gels and emulsions as early as during the cooling process. The formation of the spatial network in such systems was initiated by connections between fragments of various macromolecules, which were adjacent to each other in the solution.

Earlier measurements of the conservative module $G'$ (Figure 1) showed that intramolecular spiralisation stiffens the polymer chains in gels and emulsions already at the initial stage of starch retrogradation, during the cooling of starch solutions. The presence of fat during the gelatinisation and then the gelation of starch results in the formation of amylose complexes with fatty acids in the form of single helices [30,31]. Similarly, non-complexed ligands are trapped in the lamella [30].

Amylopectin, which is the second constituent of starch, is only slightly involved in the development of complexes. This is because the short-chain branches are not long enough to form a complex with fatty acids. Furthermore, the formation of a complex is prevented by steric hindrance between the branches of amylopectin [32].

Table 2 shows the CI (complex index) based on the decrease in the iodine binding capacity of amylose. The CI value increases along with the decrease in the starch content in the systems (the water and fat content in all systems is the same).

**Table 2.** The CI (%) values in the starch–fat emulsions at different starch concentrations (g/g) in the systems. The values are expressed as means $\pm$ SD ($n = 3$).

| $c_s$, g/g | CI, % |
|---|---|
| 0.20 | $16.03 \pm 0.79$ |
| 0.17 | $21.40 \pm 0.78$ |
| 0.13 | $32.28 \pm 0.80$ |
| 0.11 | $35.90 \pm 0.81$ |
| 0.09 | $39.30 \pm 0.82$ |
| 0.05 | $42.21 \pm 1.24$ |

Bovine fat mostly consists of long-chain fatty acids (C16, C17, and C18). Based on the results from chromatographic investigations (Figure 4 and Table 3), saturated fatty acids (C16, C17, and C18) as well as unsaturated fatty acids (C16:1, C18:1, and C18:2) are prevailing constituents. Their respective shares are approximately 49.3% and 45.8% of all fat acids.

The three aforementioned fatty acids had the greatest influence on the rheo-mechanical properties of the emulsions [33,34].

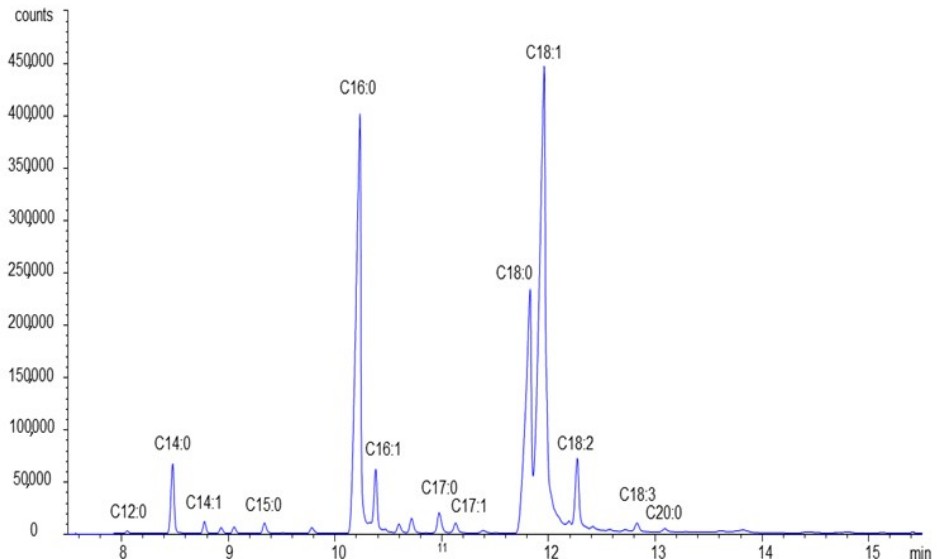

**Figure 4.** Chromatogram of examined fats (bovine fat).

**Table 3.** The percentage content of fatty acids in the fats under analysis (bovine fat).

| Fatty Acids | Beef Fat, % |
|---|---|
| C 10:0 | 0.043 |
| C 12:0 | 0.058 |
| C 13:0 | 0.012 |
| C 14:0 | 2.622 |
| C 14:1 | 0.423 |
| C 15:0 | 0.499 |
| C 16:0 | 26.493 |
| C 16:1 | 2.760 |
| C 17:0 | 1.382 |
| C 18:0 | 21.405 |
| C 18:1 | 38.734 |
| C 18:2 | 4.276 |
| C 18:3 | 0.678 |
| C 20:0 | 0.319 |
| C 20:1 | 0.289 |

The number of starch–lipid complexes is influenced by various factors, such as the cooking time and the amount and type of fatty acid added. Saturated fatty acids (12–20 carbons) form the best complexes. The complexation of C:18 unsaturated fatty acids depends on their transformation (cis or trans) [33,34].

It is impossible to clearly determine which fatty acid had the greatest share in the formation of complexes with amylose because there was a great variety of fatty acids in the fats.

Similar to the structure of starch gels, the final spatial structure of starch–fat gels undergoes reorganisation due to progressive retrogradation. In consequence, the degree of its cross-linking increases. One of the causes of this effect is the formation of new network nodes as a result of connections between adjacent macromolecular segments. Another equally important cause is the expansion of the existing nodes of the network, which consists of the binding of new segments of macromolecules on initially tetrafunctional nodes. As a result, the average functionality of the set of nodes stabilising their spatial arrangement increases. The spatial network of starch–fat gels is composed of segments formed by complex associations of amylose chains and by amylose–fat complexes. Their structure is composed of the interpenetrating networks formed by these two types of segments. As a result, in comparison with analogous starch gels, starch–fat gels are

characterised by a greater equilibrium concentration of network segments and a greater value of the functionality of final nodes $f_k$.

Like starch gels, starch–fat gels (emulsions) form dispersions. The continuous phase consists of the spatial network formed by amylose and amylose–fat complexes. The dispersed phase consists of swollen, pasted granules composed of amylopectin with un-complexed fat. In contrast to starch gels, the mechanical properties of starch–fat gels are influenced not only by the continuous phase but also by the dispersed phase.

The continuous phase determines the elastic properties, whereas the dispersed phase determines the dissipative properties of the systems under analysis. The elastic reaction of the spatial matrix creating the continuous phase is the result of the reaction of its structural elements to dynamic mechanical deformation. The spatial structures formed by elastic amylose associations are more susceptible to mechanical impact than rigid amylose–fatty complexes.

The organisation of the dispersion structure of starch–fat systems may be characteristic of both an emulsion and a gel. Basically, the following two structural forms can be distinguished: an emulsion-filled gel and a gel-stabilised emulsion.

An emulsion-filled starch gel is a spatial system in which emulsion drops are embedded. Its rheo-mechanical properties are mainly determined by the properties of the spatial matrix.

An emulsion-stabilising gel is a type of suspended gel. Its rheo-mechanical properties are mainly determined by the emulsion network.

These two types of structure are an idealisation. In fact, the structure of a particular starch–fat–water system is a combination of these two extreme forms. This means that its composite structure can be considered a hybrid network, consisting of a combination of cross-linked biopolymer molecules and partially aggregated emulsion droplets [35,36].

The rheo-mechanical properties of filled gels depend on the nature of the interaction between the filler particles and the spatial matrix [37,38]. The filler molecules can be 'active' or 'inactive' [39] or they can be 'bound' or 'unbound' [40].

The active (bound) filler molecules are physically attached to the spatial matrix of the gel and increase its strength. On the other hand, the inactive (unbound) filler molecules have a low chemical affinity for the gel network. Therefore, they behave like small holes in a network of composite materials: the value of the modulus of elasticity monotonically decreases as the concentration of particles (the hole diameter) increases [36,37].

Among the starch–fat gels under analysis, the structure of the systems with a starch concentration of $c_s = 0.17 \div 0.20$ g/g was the closest to the spatial organisation of the filled gel. This form of organisation of the structure of these systems can be justified as follows. Firstly, thanks to the presence of fats, the tested starch–fat gels were characterised by higher values of the modulus of elasticity $G'$ than the analogous starch gels. The interaction between the molecules of the filler (fat) and the gel matrix indicates the active nature of fat. It is manifested by the participation of amylose–fatty complexes in the formation of the spatial network. Secondly, the morphological structure of the nodes in the gel spatial network in the starch–fat gels was very similar to the network formed in the corresponding starch gels. This fact was confirmed by the unchanged values of the exponents of time m ($m \cong 1$), which were determined by fitting Equation (4) to the data obtained with the DMA method.

The other extreme form of organisation of the spatial structure in the systems under analysis is the gel-stabilised emulsion. This form of structural organisation can be attributed to emulsion systems with a low starch concentration, i.e., $c_s = 0.05 \div 0.11$ g/g. Amylose aggregates organised in the form of a random coil are the dominant structural elements in these gels. This fact is confirmed by the exponent of time $m \cong 3$. These systems are characterised by a lower degree of cross-linking, which is reflected by both the mean values of the final functionality $f_k$ (Table 1) and the density of segments in the $n_{ws}$ network (Figure 3).

As a result, systems with this structure have low values of the modulus of elasticity $G'$ (Figure 1) due to the fact that emulsified fat significantly determines their rheo-mechanical properties.

On the other hand, systems with an average starch content, i.e., $c_s = 0.13$ g/g can be attributed a mixed form of spatial structure organisation.

At this starch content in the systems, the rheo-mechanical properties are determined by the elastic reaction of the spatial amylose matrix together with the structures formed by amylose–fatty complexes and by non-complexed fatty acids. The exponent of time in these systems is m $\cong$ 2. This suggests that the associations of starch chains are highly ordered. The value of this exponent shows that the structures of amylose–fat complexes and non-complexed fatty acids have a lamellar order [38,39].

The physical stability of emulsions during storage is a very important problem. Emulsions are often characterised by colloidal instability. O/W emulsions have a particularly high tendency to become thermodynamically destabilised because the interfacial tension is high, and the system tends to lower it (by separation of the oil and water phases). Flocculation and coalescence are typical phenomena indicating the instability of emulsion. Flocculation is the formation of large clusters of particles (particle aggregation), but each particle retains its film (shell) [6]. Flocculation may lead to coalescence—the merging of dispersed phase particles—and thus the formation of new larger single particles. The quality and stability of the emulsion are largely determined by the size of fat particles and their distribution in the emulsion. These characteristics of the emulsion are influenced by several factors, especially by the homogenisation parameters as well as the properties of the aqueous phase (its composition and viscosity) [40]. Temperature is another important factor influencing the stability of the emulsion because it increases the interfacial tension. As a result, the amount of the emulsifier used becomes insufficient to stabilise the system [41]. According to Tadros [42], centrifugal tests based on centrifugal force are used as basic methods of assessment of the stability of a dispersion system.

The experiments (Table 4) showed that after centrifugation the animal-(beef)-fat-in-water emulsions with starch content 0.05 g/g ÷ 0.13 g/g were separated into two fractions: water and fat. These systems are gel-stabilized emulsions or have a mixed form of organization of the spatial structure. After centrifugation, no free water was found in the emulsions with higher starch content ($c_s \geq 0.17$ g/g), but a layer of released fat was observed. Its volume was greater than in the gel-stabilised emulsions.

**Table 4.** Thermal stability (SE) of model emulsions and percentage content of water and fats after centrifugation. The values are expressed as means $\pm$ SD (*n* = 3).

| $c_s$, g/g | SE, % | Vo, % | $V_{H20}$, % |
|---|---|---|---|
| 0.20 | 98.4 $\pm$ 0.79 | 100.0 | 0 |
| 0.17 | 97.20 $\pm$ 0.78 | 100.0 | 0 |
| 0.13 | 95.12 $\pm$ 0.80 | 56.0 | 44 |
| 0.11 | 94.19 $\pm$ 0.71 | 47.0 | 53 |
| 0.09 | 92.41 $\pm$ 0.69 | 35.0 | 65 |
| 0.05 | 91.34 $\pm$ 1.11 | 31.5 | 68.5 |

This means that potato starch is an emulsifier ensuring physical stability if the concentration of starch in water is at least 0.17 g/g—with a water fat content of 25%. This corresponds to starch–fat gels with the spatial structure organisation in the form of a filled gel. Emulsion-filled gels are structurally more stable than gel-stabilised emulsions because there is an active interaction between the molecules of the filler (fat) and the gel matrix. This is a consequence of the participation of amylose–fat complexes in the formation of the spatial lattice. As a result, there are higher values of equilibrium $G'$ in starch–fat emulsions than in analogous starch gels (Figure 1).

A greater share of the volume of the centrifuged fat should be associated with the lower CI value (complex index) (Table 2).

Changes in the rheo-mechanical properties of starch–fat gels, resulting from retrogradation, are mainly caused by conformational changes occurring within their structural elements, conditioned by both the concentration of starch and its physicochemical properties. Fat, which is an integral component of starch–fat gels, also undergoes transformation over time (Figure 5).

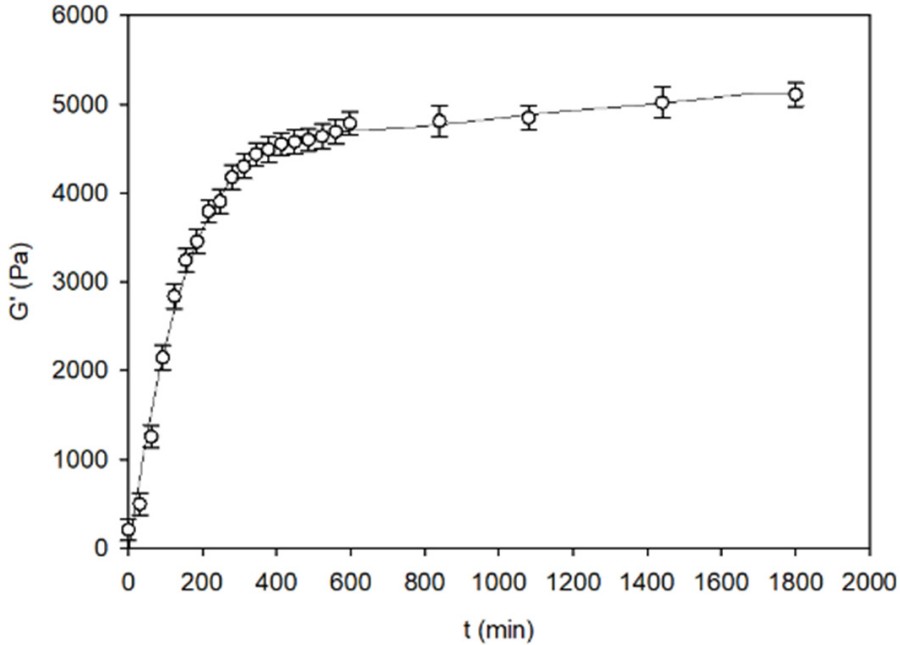

**Figure 5.** The kinetics of the storage modulus (G′) during the beef fat formation process. The values are expressed as means ± SD (*n* = 3).

During crystallisation, the fatty acids which are not associated with amylose as complexes [43,44] form a spatial network. It is not stable and is constantly changing due to the structural 'gaps' resulting from the irregular shape of fatty acid chains. This causes conformational changes in the crystal network, as a result of which fat molecules tend to achieve the most stable configuration. It is a long-term process, going beyond the time framework of the tests conducted on starch–fat gels. Therefore, it is not possible to clearly determine to what extent the fat crystallisation process affects the rheo-mechanical properties of the tested systems. Nevertheless, the presence of fats themselves significantly influences the rheo-mechanical properties of the starch–fat gels. This influence is dissipative. It is particularly noticeable in systems with a spatial structure organisation such as that of gel-stabilised emulsion.

In comparison with starch gels, the spatial systems in starch–fat emulsions are characterised by a greater density of network segments and comparable values of final functionalities $f_k$ of nodes (Table 1). Nevertheless, the morphological node structure in both systems is similar—they are composed of complex aggregates of starch chains. Despite this, both the earlier developed and newly formed segments of the spatial network in emulsions are distinguished by lower binding energy than starch gels, as evidenced by their greater capability to dissipate energy tgδ (Figure 6).

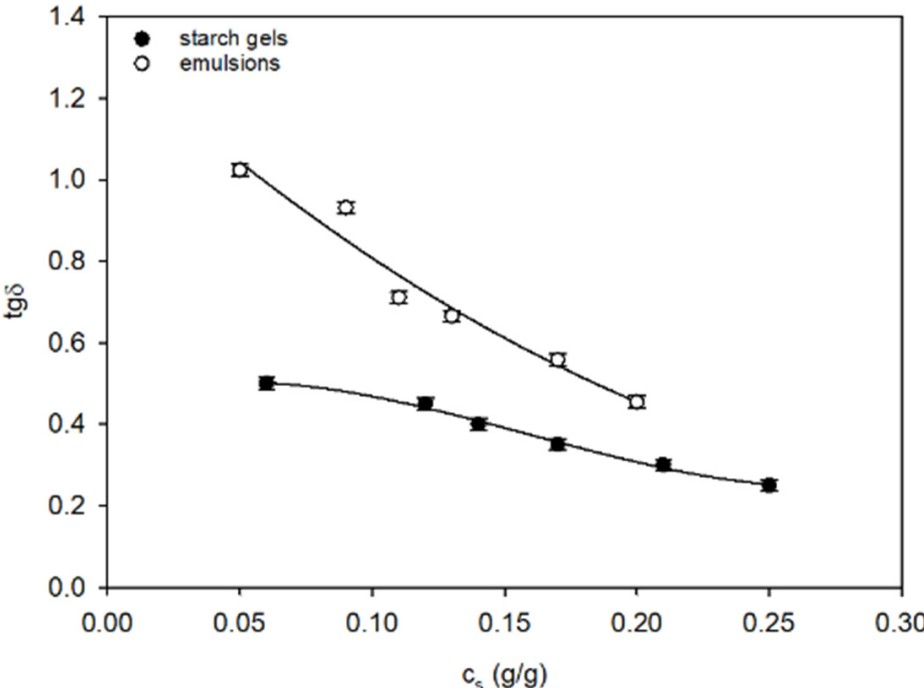

**Figure 6.** The tgδ in the process of formation of the starch gels and starch–fat emulsions. The values are expressed as means ± SD (*n* = 3).

## 4. Conclusions

The kinetics of the basic parameters determining the influence of the spatial structure of the starch–water–bovine fat systems on their rheo-mechanical properties were analysed.

The research showed that the rheo-mechanical properties of starch gels and starch–fat gels result from the conformational changes occurring within the structural elements of their spatial network. In starch gels, segments formed by complex associations of amylose chains are structural elements, whereas in starch–fat gels (emulsions) these are additionally amylose–fat complexes. Changes occurring during progressive retrogradation increase the degree of cross-linking in them. In starch gels, they are conditioned by the starch concentration, whereas in emulsions they are conditioned by the concentration of starch and the presence of fat. In consequence, the emulsions were characterised by the higher equilibrium concentration of network segments and higher mean values of the functionality of the final nodes $f_k$ than the analogous starch gels. The parameters obtained by adjusting the Avrami equation to the data obtained with the DMA method enabled the determination of three forms of organisation of the dispersion structure of starch–fat systems: emulsion-filled gel, gel-stabilised emulsions, and mixed emulsions. Each of these forms of structure organisation is conditioned by the concentration of starch in the emulsion system. The observations of the stability of the fat-in-water emulsions showed that potato starch could be an emulsifier in such systems. However, the starch content in systems containing 25% fat in water should not be lower than 0.17 g/g in relation to water.

**Funding:** This research received no external funding.

**Institutional Review Board Statement:** Not applicable.

**Data Availability Statement:** Not applicable.

**Conflicts of Interest:** The author declares no conflict of interest.

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
