# Peer review of "The Role of Starch in Shaping the Rheo-Mechanical Properties of Fat-in-Water Emulsions"

_2673-4176, doi:10.3390/polysaccharides3040047_

Round 1

Reviewer 1 Report

This study is aiming to produce some interesting investigations about the impact of the molecular structure of bovine fat-in-water type emulsions stabilised by potato starch on their rheomechanical properties. I think it is a valuable work. The authors can further consider and clarify the correlation between the emulsion structure and the rheomechanical properties.  Can the author provide some data or pictures to illustrate the relationship between the two. I think the reason why materials affect macroscopic properties, like rheomechanical properties, may be that they affect microscopic structure in the first place. The accuracy of English writing needs further improvement.

Author Response

Rev.1

Comments and Suggestions for Authors

This study is aiming to produce some interesting investigations about the impact of the molecular structure of bovine fat-in-water type emulsions stabilised by potato starch on their rheomechanical properties. I think it is a valuable work. The authors can further consider and clarify the correlation between the emulsion structure and the rheomechanical properties.  Can the author provide some data or pictures to illustrate the relationship between the two. I think the reason why materials affect macroscopic properties, like rheomechanical properties, may be that they affect microscopic structure in the first place. The accuracy of English writing needs further improvement.

I regret to say that at the moment I do not have any images illustrating the relationship between the microscopic structure and the rheological properties of the emulsion.

I and my team made a computer image analysis to illustrate the distribution of fat in fat-water emulsions emulsified with potato starch. We examined the structure of the preparations under a constant magnification of the microscope (x100). We analysed ten fields with fixed areas for each preparation. At the moment we are processing the images. The characterisation of the images is based on the area of fat fields and the percentage of fat fields in the field of view. Please, find attached two images of starch-fat emulsions with starch content: cs = 0.17 g/g and cs = 0.05 g/g as examples. Computer image analysis with tests based on NMR and texture analysis will be the subject of our next study.

Reviewer 2 Report

This article is devoted to the role of starch in the formation of the properties of fat-in-water emulsions. The article is well written and easy to understand. The main provisions are clearly articulated. Obtaining systems of the "fat-in-water" type is interesting and has practical significance, as it is actively used in various fields. Despite the many advantages of this article, there are some points that it is desirable to finalize:

1. Fat-in-water systems have been studied for a long time. in the introduction, please add where they apply in more detail.

2. Figure 1. If it's about kinetics, has the rate/constants been calculated? At least according to the initial data (up to 600 min.).

3. In general, more discussions and comparisons with the literature can be added to the article.

4. What polysaccharides other than starch were used in such a system?

5. Please cite work: 10.3390/foods10112571

Author Response

Rev.2

Comments and Suggestions for Authors

This article is devoted to the role of starch in the formation of the properties of fat-in-water emulsions. The article is well written and easy to understand. The main provisions are clearly articulated. Obtaining systems of the "fat-in-water" type is interesting and has practical significance, as it is actively used in various fields. Despite the many advantages of this article, there are some points that it is desirable to finalize:

  1. Fat-in-water systems have been studied for a long time. in the introduction, please add where they apply in more detail.

The Introduction has been corrected and supplemented with the required information according to the Reviewer’s suggestions.

  1. Figure 1. If it's about kinetics, has the rate/constants been calculated? At least according to the initial data (up to 600 min.).

Figure 1 shows the kinetics of changes in storage modulus G’. These data were obtained directly in the measurements. The nature of the course of dependence G’(t) and the kinetics of network segment density is identical due to the relationship between G’ and ns (ns = G’/RT). The manuscript does not have a figure showing kinetics ns(t) but the parameters resulting from fitting the data to the equation ns = nos + [nws – nos] {1 – exp[– (kt)m]}. The discussion has been supplemented with the initial mean G’o values for gels and emulsions. The final (equilibrium) values are shown in Figure 1. Kinetics G’(t) follows the equation analogous to the equation for ns: G’(t) = G’o + [G’ws – G’o] {1– exp[– (kt)m]}. The m values are the same for G’ and ns, and the course of dependence k(cs) is similar for G’ and ns.

  1. In general, more discussions and comparisons with the literature can be added to the article.

            The Discussion has been supplemented with a paragraph on the stability of the emulsions under analysis.

  1. What polysaccharides other than starch were used in such a system?

            Only native potato starch and chemically and physically modified potato starches were used in the research. Our study presents only the results of rheological measurements for native starch.

  1. Please cite work: 10.3390/foods10112571

            This reference publication was cited in the Introduction.

Reviewer 3 Report

The author reported a study where it is presented the role of starch with the fat-based emulsions. The author presented mainly rheological aspects and study of the composition. The presented results supported the expected outcome; however, there are aspects that they were not fully understood, and it need further clarification.

Please find below some suggestions/comments that might help in improving the quality of the manuscript:

1.      The author selected as material to shape the rheomechanical properties of the fat-based emulsions, starch but not explanation why they have stopped only at this biopolymer; the authors are requested to insert some arguments and their selection criteria.

2.     The author presented GC MS results, but no chromatogram inserted; the authors are requested insert a chromatogram where from they calculated the composition.

3.      The purpose of the study was not clear enough. If any application demands this adjustment with starch, the author is advised to mention it; the author is advised to reformulate the phrase with the purpose of the study.

4.      The author mentioned about starch concentration, but no optimal value was mentioned. The author is requested to conclude on an optimal concentration.

Author Response

Rev.3

Comments and Suggestions for Authors

The author reported a study where it is presented the role of starch with the fat-based emulsions. The author presented mainly rheological aspects and study of the composition. The presented results supported the expected outcome; however, there are aspects that they were not fully understood, and it need further clarification.

Please find below some suggestions/comments that might help in improving the quality of the manuscript:

  1. The author selected as material to shape the rheomechanical properties of the fat-based emulsions, starch but not explanation why they have stopped only at this biopolymer; the authors are requested to insert some arguments and their selection criteria.

It is true that starch is not the only water-binding hydrocolloid with structure-forming properties. There are many other hydrocolloid preparations based on proteins and other polysaccharides – mainly gums (guar gum, gum acacia, xanthan gum, gellan gum) and pectins. These hydrocolloids are used in the food industry mainly for the production of fruit preserves (jams, jellies), vegetable preserves, and juices.

            Why starch? First of all, in Poland starches are mainly used as a fat substitute in cold cuts. Second of all, it is the most readily available hydrocolloid obtained from common crops (various cereals) or tubers (potatoes, sweet potatoes). Thanks to this, the cost of starch production is much lower. Third of all, the physicochemical properties of starch can be widely modified. As a result, it is possible to obtain a hydrocolloid tailored to its function in the product (structuring agent or fat substitute). Fourth of all, starch, especially its amylose fraction, is capable of forming complexes with various organic and inorganic compounds (including iodine, alcohols, free fatty acids, emulsifiers, monoacylglycerols and other surfactants). Thanks to this, starch can be used as an emulsifier in various emulsions.

  1. The author presented GC MS results, but no chromatogram inserted; the authors are requested insert a chromatogram where from they calculated the composition.

            A chromatogram can be found in the manuscript.

  1. The purpose of the study was not clear enough. If any application demands this adjustment with starch, the author is advised to mention it; the author is advised to reformulate the phrase with the purpose of the study.

The aim of the study has been corrected.

  1. The author mentioned about starch concentration, but no optimal value was mentioned. The author is requested to conclude on an optimal concentration.

            The Discussion has been supplemented with information on the stability of starch-fat emulsions. The observations of the stability of the fat-in-water emulsions showed that potato starch could be an emulsifier in such systems. However, the starch content in systems containing 25% fat in water should not be lower than 0.17 g/g in relation to water.

Round 2

Reviewer 2 Report

accepted

Reviewer 3 Report

The authors answered to the addressed queries and the manuscript was updated accordingly.